# Inhibitory Activity of Essential Oils of *Mentha spicata* and *Eucalyptus globulus* on Biofilms of *Streptococcus mutans* in an In Vitro Model

**DOI:** 10.3390/antibiotics12020369

**Published:** 2023-02-10

**Authors:** Guillermo Ernesto Landeo-Villanueva, María Elena Salazar-Salvatierra, Julio Reynaldo Ruiz-Quiroz, Noemi Zuta-Arriola, Benjamín Jarama-Soto, Oscar Herrera-Calderon, Josefa Bertha Pari-Olarte, Eddie Loyola-Gonzales

**Affiliations:** 1Faculty of Pharmacy and Biochemistry, Universidad Nacional Mayor de San Marcos, Jr. Puno 1002, Lima 15001, Peru; 2Institute for Research in Biological Chemistry, Microbiology and Biotechnology “Marco Antonio Garrido Malo”, Faculty of Pharmacy and Biochemistry, Universidad Nacional Mayor de San Marcos, Lima 15001, Peru; 3Faculty of Health Science, Universidad Nacional del Callao, Av. Juan Pablo II No. 306, Bellavista, Callao 07011, Peru; 4School of Human Medicine, Faculty of Health Sciences, Universidad Peruana Unión, km 19 Carretera Central, Ñaña, Lurigancho Lima 15457, Peru; 5Department of Pharmacology, Bromatology and Toxicology, Faculty of Pharmacy and Biochemistry, Universidad Nacional Mayor de San Marcos, Jr. Puno 1002, Lima 15001, Peru; 6Department of Pharmaceutical Chemistry, Faculty of Pharmacy and Biochemistry, Universidad Nacional San Luis Gonzaga, Ica 11001, Peru; 7Department of Pharmaceutical Science, Faculty of Pharmacy and Biochemistry, Universidad Nacional San Luis Gonzaga, Ica 11001, Peru

**Keywords:** medicinal plants, dentistry, essential oils, antibiofilm, volatile oil, biofilms, volatile oils

## Abstract

The aim of this study was to evaluate the inhibitory activity of the commercially available essential oils of *Mentha spicata* (spearmint) and *Eucalyptus globulus* (eucalyptus) on *Streptococcus mutans* ATCC 25175 biofilms in vitro, emulating dental plaque conditions. The composition of the essential oils (EOs) was determined using gas chromatography coupled with mass spectrometry (GC-MS), with the main metabolites being Carvone (57.93%) and Limonene (12.91%) for *Mentha spicata* and 1,8-Cineole (Eucalyptol) (65.83%) for *Eucalyptus globulus*. The inhibitory activity was evaluated using the methods of agar-well diffusion and colorimetric microdilution. The inhibition halos were 18.3 ± 0.47 mm and 27.0 ± 0.82 mm, and the MICs were 1.8484 mg/mL and 1.9168 mg/mL for the EOs of *Mentha spicata* and *Eucalyptus globulus,* respectively. The activity against the biofilms was evaluated on a substrate of bovine enamel pieces using a basal mucin medium (BMM) in anaerobic conditions with daily sucrose exposition cycles in order to emulate oral cavity conditions. The EOs were applied in a concentration of 0.5% in a sterile saline vehicle with 1% polysorbate 20. After 72 h of cultivation, a significant reduction was observed (*p* < 0.001%) on the biofilm biomass, which was evaluated by its turbidity in suspension and using a count of the recoverable organisms with regards to the control. The effects of the Eos were not significantly distinct from each other. The EOs showed antimicrobial activity against both the Streptococcus mutans planktonic and biofilm cultures. Thus, EOs may have great potential for the development of pharmaceutical and sanitary products for oral health.

## 1. Introduction

The human oral cavity is a warm, moist and nutrient-rich environment. It supports a wide range of microorganisms with around 700 prokaryote species (54% are officially named, 14% are unnamed but cultivated and 32% are known only as uncultivated phylotypes) [1]. These organisms inhabit the oral cavity, forming biofilms on all the available surfaces. The relative composition and general structure of these are relatively specific to each area and depend on the local conditions [2]. They are a normal part of a healthy mouth and allow for the normal function of a host’s physiology and defence [3]; for instance, by offering resistance to the colonization of exogenous and potentially pathogenic organisms [4]. The hard non-shedding surface of the tooth allows the development of a very complex dental biofilm with an exceptionally high cell density. Inside the biofilm, both commensal organisms and opportunistic pathogens live in a dynamic equilibrium within the host, its disruption could lead to pathological processes [5,6]. On the other hand, cavities are a product of one of these disruptions. It is concretely attributed to an increase in acid uric and acidogenic bacteria in the dental biofilm. The organisms of the sub-group mutans-Streptococci (which include *Streptococcus mutans*) are considered the main etiological agents of dental caries. These organisms use fermentable carbohydrates of dietary origin as a primary energy source; the by-products of this process reduce the pH and may cause irreversible deteriorate to teeth [7,8].

The use of in vitro models to study the potential effects of natural products based on essential oils from aromatic plants are currently used to screen for the antibiofilm effect against several bacteria, which are the main cause of caries [9]. There are several approaches to emulate the oral conditions, but methodological differences, and the use of mono- or multi species-cultures make it difficult to compare those results. [10]. On the other hand, *Streptococcus mutans* has been implicated as the main cause of dental caries in humans, having the capacity to create a biofilm on tooth surfaces known as dental plaque, which is one of the proposed mechanisms due to the activity of glucosyltransferases (GTFs). The bacteria synthesize glucan from sucrose and glucans, which subsequently facilitates the solid adhesion of its cells to tooth surfaces [11].

*Mentha spicata (M. spicata*) is an aromatic plant belonging to the Lamiaceae family. The leaves are popularly used as a tea flavouring agent and the whole plant is used as a carminative. The fresh and dried plants and their essential oils are widely used in the food, cosmetic, confectionary, chewing gum, toothpaste, and pharmaceutical industries [12]. *M. spicata* possesses several biological activities and is used in folkloric medicine as a carminative, antispasmodic, diuretic, antibacterial, antifungal, antioxidant agent, for the treatment of colds and flu, respiratory tract problems, gastric pain, haemorrhoids, and stomach-aches [13,14]. On the other hand, *Eucalyptus globulus* (*E. globulus*) from the Myrtaceae family is used as an antibacterial agent against *P. gingivalis* and *S. mutans*, which are linked to periodontitis [15]. Therefore, its main component 1,8-cineneol from the essential oil is used in several products, such as mouthwash i.e., Listerine (Pfizer) and toothpaste for oral hygiene [16].

Extensive evidence supports the antimicrobial properties of essential oils against oral and non-oral bacteria [17]. Nevertheless, most of the studies are limited to planktonic culture models and only a few consider biofilm growth conditions. There is increasing evidence to support its use as coadjutants of oral hygiene [18]. Therefore, the present study aimed to evaluate the antimicrobial activity of commercially available essential oils of *Eucalyptus globulus* and *Mentha spicata* against *S. mutans* biofilms using an in vitro model.

## 2. Results

### 2.1. Chemical Analysis by GC-MS of the Essential Oils of M. spicata and E. globulus

The analysis of the EOs of M spicata revealed 19 volatile compounds with carvone being the main chemical component at 57.93%, followed by Limonene at 12.91% (Table 1). Regarding the EOs of *E. globulus*, the main components were 1,8-Cineol (eucalyptol) at 65.83% and α-Pinene at 18.15% as are shown in Table 2.

### 2.2. Antibacterial Activity of the Essential Oil of M. spicata and E. globulus against S. mutans

According to our results, based on the evaluation of the ability to form biofilms using the Congo Red method, punctate, dry, and black colonies were observed. Clearly showing that the black pigmentation fades throughout the agar medium (Appendix A). This supports the ability of *S. mutans* ATCC 25,175 to form biofilms on teeth.

The inhibition halos presented by the essential oils can be seen in Table 3 and Figure 1, which noted that the maximum zone inhibition was at 100% of the concentration in both the EOs, followed by a 50% of concentration (*v*/*v*). On the other hand, the minimum inhibitory concentration (MIC) of the essential oils against *S. mutans* ATCC 25,175 for *M. spicata* was 1.8484 mg/mL and for *E. globulus* was 1.917 mg/mL.

### 2.3. Antibiofilm Activity Using an In Vitro Model on Dental Enamel Pieces of Bovine Origin

#### 2.3.1. Culture Medium pH Fluctuation

The different experimental groups showed fluctuating levels of acidification in the culture medium during this study, these are shown in Figure 2. The positive control group showed significantly higher acidification in comparison to any of the other groups. On the other hand, the negative control group exhibited no major changes. Initially, the treatment groups exhibited lesser acidification than the positive control groups. Nevertheless, this effect was lost as the culture continued. Although the capability of the *E. globulus* EO to reduce acidification showed slightly better results, no significant difference was observed between the effects of each essential oil in this parameter.

#### 2.3.2. Spectrophotometric Turbidity

The total biofilm biomass was indirectly assessed through the comparison of the spectrophotometric turbidity of the final bacterial suspension of the enamel slabs in the different experimental groups (Figure 3).

The letters indicate which groups had a statistically significant difference with a confidence interval above 95%. There was no significant difference between the two treatment groups, neither was there one between the negative control group and the EO *E. globulus* group. However, there was a clear and significant difference between the positive control group, the negative control, and both treatment groups.

#### 2.3.3. Recoverable Microorganism Count

The count of the microorganisms recoverable from the final bacterial suspension in each of the enamel slabs (Figure 4).

There is no significant difference between the two treatment groups, regardless both were significantly different from both the positive and negative control groups. The mean of the bacterial counts from the positive media group was in the order of 14.25 × 10^7^ CFU/piece; the treatment groups had a mean of 5.56 × 10^7^ CFU/piece and 8.42 × 10^7^ CFU/piece for *E. globulus* and *M. spicata,* respectively; and the negative control group had a mean of 6.08 × 10^4^ CFU/piece, nevertheless most of the slabs had no recoverable microorganisms.

## 3. Discussion

The results of the GC-MS analysis revealed 19 compounds for *M. spicata* 19 with the most abundant being Carvone (57.93%) and Limonene (12.907%). Both monoterpenes are, although in variable proportions, consistently reported as the most abundant constituents of this essential oil in the literature. Snoussi et al. [19] reported concentrations of 40.8% ± 1.23% and 20.8% ± 1.12%, and Aggarwal et al. [20] reported 56.6% and 27.3% for Carvone and Limonene, respectively. Likewise, these are the components to which the antimicrobial activity of the essential oil of *M. spicata* is attributed, both in planktonic and biofilm models.

The essential oil of commercial origin from *E. globulus* identified nine compounds, with the most abundant being 1,8-Cineole (eucalyptol) (65.830%) and α-Pinene (18.150%). Goldbeck et al. reported concentrations of 71.05% and 8.30%, and Song et al. reported concentrations of 72.71% and 9.22% for 1,8-Cineole (eucalyptol) and α-Pinene, respectively. The antimicrobial activity of the essential oils from plants of the genus Eucalyptus is normally attributed to the concentration of 1,8-Cineole. In comparative studies of the activity of the essential oils of *E. globulus* and *E. urograndis* against *Streptococcus mutans*, the main phytochemical difference between the plant species was the higher concentration of 1,8-Cineol in *E. globulus* (71.05%) compared to *E. urograndis* (36.18%) [21].

The Congo red dye is a non-selective dye used to detect the exopolysaccharides of the biofilm structure. The results obtained demonstrate similar results to those reported by Alghamdi [22] and Jain et al. [23] for the isolation and characterization of biofilm-forming microorganisms from the oral microbiota. The inhibition halos observed in the agar-well diffusion test demonstrate a significant inhibitory activity of the essential oils of *M. spicata* and *E. globulus* against *Streptococcus mutans* ATCC 25175. The bibliography reports varied diameters for the inhibition halos for this test; Rasooli et al. [24] reported for the essential oils of *M. spicata* and *E. camaldulensis* halos of 60 ± 0 mm and 47 ± 2.65 mm, respectively, against *S. mutans* PTCC 1601; Goldbeck et al. [21] reported a diameter of 34.7 ± 0.6 mm for the essential oil of *E. globulus* against *S. mutans* ATCC 700610; in contrast Kumar et al. [25] reported a diameter of 3.44 ± 0.37 and 0.0 mm for the essential oils of *E. globulus* and *M. spicata,* respectively, against *Streptococcus mutans* ATCC 25175. When comparing these widely varied data we must take into account both the biological and methodological considerations. The former responds to the variable nature of the composition of the essential oils in response to their cultivation conditions (understood as edaphological factors or those referring to the cultivation methods for commercial purposes, and the technological investment in improving these) to differences in the steam distillation conditions of the oil or its storage conditions, and the use of different strains of *Streptococcus mutans*. On the other hand, no less important, are the methodological considerations; Rasooli et al. [24] and Goldbeck et al. [21] used diffusion disks instead of agar wells and although both used the Mueller–Hinton medium, they do not report having supplemented with 5% sterile lamb blood, as recommended by CSLI M02-A11 for organizations with demanding requirements. It must also be considered that for the purposes of this study the volume of the inoculum was increased 2.5 times. In the case of what was reported by Kumar et al. [25], the selected medium was blood agar and its incubation time was 48 h, double that recommended by the CSLI (time used in the present study and in the rest of the bibliography), which could have significantly reduced the impact of the essential oils in the test. Dhifi et al. [26] evaluated the activity of the *M. spicata* essential oil against Gram (+), Gram (−) organisms, and fungi. Among the former, inhibition halos of 14.7 ± 0.3 mm and 12.5 ± 0.6 mm were reported for *Staphylococcus aureus* ATCC 29,213 and *Staphylococcus epidermis* A233, respectively.

The MICs were determined for the essential oils of *E. globulus* and *M. spicata*; it should be noted that the values reported for both oils are not significantly different from each other (*p* > 0.832). These results are according to Rasooli et al. [24], who reported a MIC against *Streptococcus mutans* of 2 mg/mL for both *M. spicata* and *E. camaldulensis* essential oils. A relevant observation to consider when comparing these results is that in this study a macrodilution method was used to determine the MIC. Considering the results of the microdilution test, and the diffusion in the agar well, it can be affirmed that *Streptococcus mutans* ATCC 25,175 is susceptible to inhibition by the two essential oils evaluated; however, no significant difference is observed between them.

In general, this group of aromatic compounds is considered to be responsible for the reported antimicrobial properties of the EOs, both in planktonic and biofilm models [27,28]. The exact mechanism by which these compounds wield this effect has yet to be completely elucidated. However, this may be at least partially due to the gross disruption of the lipidic fraction of the cytoplasmic membrane of the microorganism [29]. This affects not only its functions as a barrier but also its role as an enzymatic matrix or as an energy transductor. Moreover, it is not discarded that these compounds may penetrate the cell and interact with other intracellular sites critical for antibacterial activity [30].

The use of a mucin-based culture medium, the cyclical exposure of the biofilm to fermentable carbohydrates, followed by a significant reduction in its concentration (in the case of this study the feast-famine cycles), and the control environmental factors, such as the medium pH are usually considered as essential parameters to control in most in vitro biofilm models [31]. In regard to the mucin employed, not many bacteria can use this complex glycoprotein as a sole source of nitrogen and carbon. In particular, it cannot serve as a primary energy or carbon source for *S. mutans* [32]. Nevertheless, it has been demonstrated that, in the presence of a primary energy source, such as glucose, it can provide amino acids to support the growth of *S. mutans*. Moreover, it can enhance the long-term survivability of the organism during periods of starvation [32].

The fluctuation in the medium pH (Figure 2) during the experiment can be attributed to the biological activity of the *S. mutans* biofilm [33]. This is reflected in the fact that the major acidification occurred in the positive control group where the biofilm grew unmolested. Although initially the treatment moderated the medium acidification, this effect was lost as the culture continued. It could be speculated that a longer exposure time or more exposures per day could have increased the effectiveness of the treatment. However, it is important to remember that clinically, chemical treatments to control oral biofilms are considered mostly coadjutants to normal oral hygiene; the mechanical removal of the dental plaque through tooth brushing with a fluoride containing toothpaste is considered the principal mean of plaque control [33].

Both the spectrophotometric turbidity of the final bacterial suspensions (Figure 3) and the total recoverable microbial count (Figure 4) can be used to indirectly evaluate the capability of the EOs treatment to inhibit the development of the *S. mutans* biofilm. In both parameters, the inhibitory effect was statistically significant (*p* < 0.001). Although slightly better results were observed for the EOs of *E. globulus* in comparison to the EOs of *M. spicata* no statistically significant difference was observed, suggesting a similar effectiveness. This difference is consistent with the findings of Rasooli et al. [24], who tested the effects of the EOs of *M. spicata* and *E. camaldulensis* (another member of the Eucalyptus genus) both in vivo as well as in vitro. In the latter, a significant reduction in the specific biofilm formation was observed for the EOs in comparison to the control. However, the Eucalyptus’s treatment showed better results. Similar results were observed in the in vivo model [34].

## 4. Materials and Methods

### 4.1. Obtention of the Essential Oil

Essential oils of *Mentha spicata* and *Eucalyptus globulus* were purchased from the Essential Oils Peru ^®^ company (Lima, Peru).

### 4.2. Composition Analysis by Gas Chromatography-Mass Spectrometry (GC/MS) of Essential Oils

The oils were analyzed using a gas chromatograph coupled with the mass spectrometry (GC/MS) model 7890A GC system Agilent Technologies/5975C inert XL EI/CI MSD with a triple axis detector; the column used was the J&W DB-5 ms model (30 m, 0.25 mm, 0.25 µm, 7” coating) [35]. Each retention index was calculated and compared with a homologous series of n-alkanes C9–C25 (C9, BHD purity 99% and C10–C25, Fluka purity 99%). The relative amount (expressed as a percentage) of each compound identified in the EO was calculated by comparing the area of the corresponding peak in the chromatogram with the total area of identified peaks. No correction factor was applied [36].

### 4.3. Inhibitory Activity of the Essential Oils M. spicata and E. globulus and Determination of the Minimum Inhibitory Concentration

#### 4.3.1. Evaluation of the Ability to Form Biofilms by the Congo Red Method

The biofilm formation capacity of the study organism, named *Streptococcus mutans* ATCC 25,175 (KWIK-STIK TM Microbiologics^®^, St. Cloud, MN, USA), was evaluated using the Congo Red agar method as described by Freeman et al. [37]. Congo Red is a non-selective dye for matrix exopolysaccharides. After sowing, the medium was incubated at 37 °C for 24 h in aerobic conditions. The microorganisms capable of forming biofilms have black, dry colonies; those without have red colonies.

#### 4.3.2. Agar-Well Diffusion Method

The antimicrobial activity of the EOs was evaluated using a modified version of the agar-well diffusion method described by Rojas et al. [38]. The test is based on the diffusion of an antimicrobial agent in a solid medium causing the inhibition of microbial growth, the latter is evidenced by the formation of clear halos without microbial development around the substance. The diameter (in millimeters) of these halos is indicative of the antimicrobial properties of the substance analyzed.

Following the recommendations of CLSI M07-A9 [39] for microorganisms with demanding requirements, the Mueller–Hinton agar was supplemented with 5% (*v*/*v*) sterile lamb blood. Likewise, changes were made to the volume and concentration of the inoculum, and in the total volume of the agar per plate. Finally, the culture was carried out in anaerobic conditions. These last modifications did not alter the final microbial load per plate of 1.5 × 10^6^ CFU/mL and were carried out to facilitate the growth of the bacteria in response to the results obtained in the growth controls of the pilot phase.

Following 24 h, cultures of *Streptococcus mutans* ATCC 25,175 on BHI agar supplemented with 1% sucrose, a bacterial suspension was prepared in sterile 0.9% saline solution with a turbidity equivalent to 0.5 on the McFarland scale (approx. 1.5 × 10^8^ CFU/mL). Then, 1 mL of this suspension was inoculated to each 100 mL of the Mueller–Hinton agar supplemented with 5% (*v*/*v*) sterile lamb blood. The medium was inoculated at a temperature of 45 °C, homogenized and distributed in sterile 90 × 15 mm glass Petri dishes at a volume of 25 mL per plate. When the medium was solidified, 9 mm diameter wells were dug.

To 100 µL of the respective pure essential oils and their 50% and 10% dilutions in dimethylsulfoxide (DMSO) were added to the respective wells and left to settle for no less than 60 min to allow for diffusion at room temperature. Similarly, a growth control and solvent blank control plates were included. Incubation was carried out for 24 h at 37 °C under anaerobic conditions. The test was performed three times, with each dilution tested in duplicate. To facilitate the reading of the results, this was performed using a light source under the plate.

#### 4.3.3. Determination of the Minimum Inhibitory Concentration (MIC) by the Colorimetric Microdilution Method

The minimum inhibitory concentration (MIC) was determined using a modified version of the CSLI M07-A9 protocol [39], adapting the preparation of the resazurin indicator from that described in the Liu et al. [40] protocol, as well as laboratory-specific modifications regarding the EO sample preparation using Tween 20 surfactant. Brain Heart Infusion as culture media, and 5 × 10 ^5^ CFU/mL as final inoculum of *Streptococcus mutans* ATCC 25,175. The test was performed in triplicate, with three replicates per test (three equal rows per plate).

### 4.4. In Vitro Model for the Formation of Biofilms on Dental Enamel Pieces of Bovine Origin

#### 4.4.1. Culture Media

The biofilms were grown in a modified version of Basal Mucin Medium (BMM), as described by Wong et al. [41]. This culture medium was used as a human saliva analogue and contained 2.5% of type III porcine gastric mucin (Sigma Chemical Co. St Luis, MO, USA). It was supplemented with 0.1 mM glucose to resemble the basal concentration of glucose in human saliva. The pH of the medium was adjusted to 7.0 ± 0.1 with 0.1 N NaOH or 0.1 N HCl.

#### 4.4.2. Activation of the Microorganism and Preparation of the Inoculum

Streptococcus mutans ATCC 25,175 (KWIK-STIK TM Microbiologics^®^) was initially activated using BHI agar supplemented with 1% sucrose under an anaerobic environment, later it was kept in a modified version of the *Mitis salivarus* medium.

The inoculum was prepared from an overnight culture of BHI agar supplemented with 1% sucrose. A loopful of colonies were taken and suspended in a sterile saline solution; this suspension was adjusted to 4 on the McFarland scale (approx. 12 × 10^8^ CFU/mL). Then 500 µL of this suspension was diluted to up to 10 mL with BMM supplemented with 0.1 mM of glucose (approx. 6 × 10^7^ CFU/mL).

#### 4.4.3. Preparation of Enamel Slabs

Twenty-four square enamel bovine slabs (2.0 mm thickness and 4.0 mm of sides) were used as the substratum on which to anchor the biofilm grown. These were cut from the buccal side of the central incisors of recently slaughtered cows and were later polished to eliminate the dentine. Finally, they were suspended in saline solution and autoclaved.

#### 4.4.4. Biofilm Cultivation and Treatment

Biofilms were cultured in a 72 h semi-continuous batch model, using the enamel slabs as substratum and under anaerobic conditions. After an initial cultivation faze, the cultures received three feast-famine cycles of exposure to fermentable carbohydrates (emulating the dietary exposure to carbohydrates) and one treatment per day.

The protocol used was a modified version of the one described by Ccahuana–Vásquez et al. [42]. Initially, the enamel slabs were aseptically put into 24-well cellular culture plates (NUNC TM) containing 1 mL of the culture medium and incubated for 1 h at 37 °C, until the formation of a conditioning layer over the surface of the substratum. Then the medium was removed by absorption and replaced with 1 mL of either inoculum or fresh sterile medium. These were incubated in anaerobic conditions for 8 h at 37 °C, then 1 ml of fresh medium was added and the incubation continued. On the second and third days, the feast and famine cycles began three times a day.

On each of the three daily feast-famine cycles, the culture medium was removed by absorption, then 1 mL of 10% sterile sucrose was added and incubated under anaerobic conditions for 10 min after which the sucrose was removed. Then, the slabs were washed with sterile 0.01% NaCl and finally 2 mL of fresh medium was added to each well. From hour 22.5 (before the first exposure to the sucrose and treatment) to 54 (before the last exposure to sucrose), each time the culture medium was removed and kept to monitor its pH.

The EOs were used in a concentration of 0.5% in a sterile saline vehicle with 1% polysorbate 20 (autoclaved). This surfactant was selected as it showed no antimicrobial effect on *S. mutans* in a microdilution test previous to this study (data not shown). The treatments solutions were applied by immersion of the enamel slabs and maintained for 1 min once a day. Then, these were removed by absorption and the slabs were rinsed three times using a 0.9% sterile saline solution. This was after the first exposition to sucrose 10% and its washing and before the addition of fresh medium.

In this model, four experimental groups were considered: a positive control (inoculated and under treatment only with a control vehicle), a negative control (not inoculated and under treatment only with a control vehicle), and two treatment groups of *E. globulus* and *M. spicata,* respectively, (both inoculated and treated). Each group contained six enamel slabs.

#### 4.4.5. Biofilm Collection and Evaluation of the Treatment

After 72 h of incubation, the medium was retired by absorption and each measurement was made at day 2 and 3. The slabs were rinsed again and aseptically submerged in sterile vials containing 5 mL of the same solution. Then, they were sonicated for one minute at 40 MHz (Ultrasonic bath UC-20–JEIOTECH). Finally, the slabs were aseptically removed. The turbidity of the resulting suspensions was measured at 580 nm (UV-Vis Spectrophotometer Spectroquant^®^ Prove 300) to estimate the biofilms biomass. The total microbial count of recoverable microorganisms was conducted on 10^−3^ to 10^−7^ dilutions, using a standard plate count agar supplemented with 1% of 2,3,5-Triphenyl tetrazolium chloride (TTC) (Merck) incubated at 37 °C for 48 h.

### 4.5. Statistical Analysis

The differences among the groups were evaluated using a one-way analysis of variances (ANOVA), for both the turbidity and the total microbial count. The posthoc Tukey analysis was used to determine the statistical significance of the differences among the treatment groups as defined. The calculus and the graphics were made using GraphPath Prism 8 software and the significance level was considered at 5%.

## 5. Conclusions

The essential oils of commercial origin of *Mentha spicata* (mint) and *Eucalyptus globulus* (eucalyptus), under the conditions worked, reduced the biomass of the biofilms of *Streptococcus mutans* ATCC 25,175 when evaluated by its turbidity in suspension and by the count of the recoverable microorganisms.

## Figures and Tables

**Figure 1 antibiotics-12-00369-f001:**
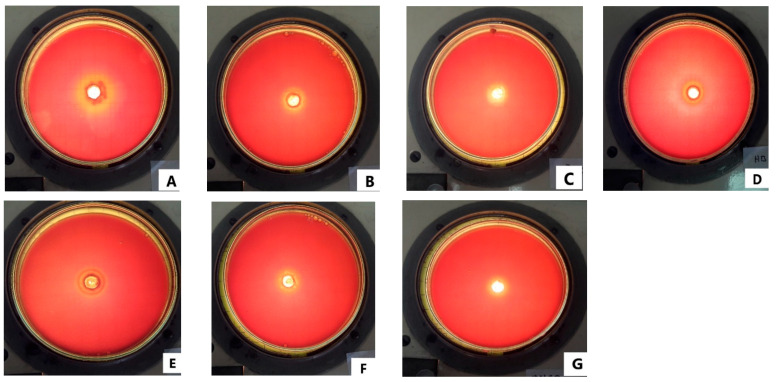
Agar-well diffusion method. (**A**) *E. globulus* EO 100% (**B**) *E. globulus* EO 50% (**C**) *E. globulus* EO 10% (**D**) *M. spicata* EO 100% (**E**) *M. spicata* EO 50% (**F**) *M. spicata* EO 10% (**G**) Solvent (DMSO 0.1%).

**Figure 2 antibiotics-12-00369-f002:**
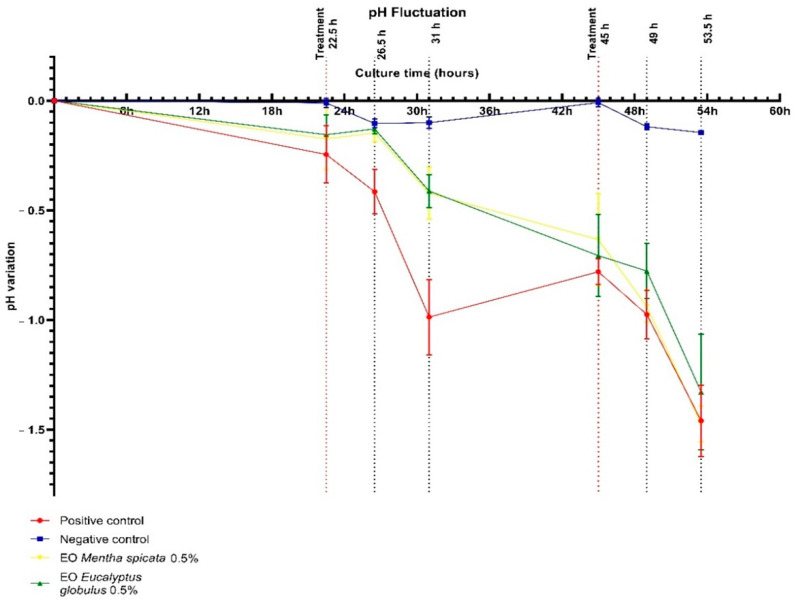
pH fluctuation in culture medium.

**Figure 3 antibiotics-12-00369-f003:**
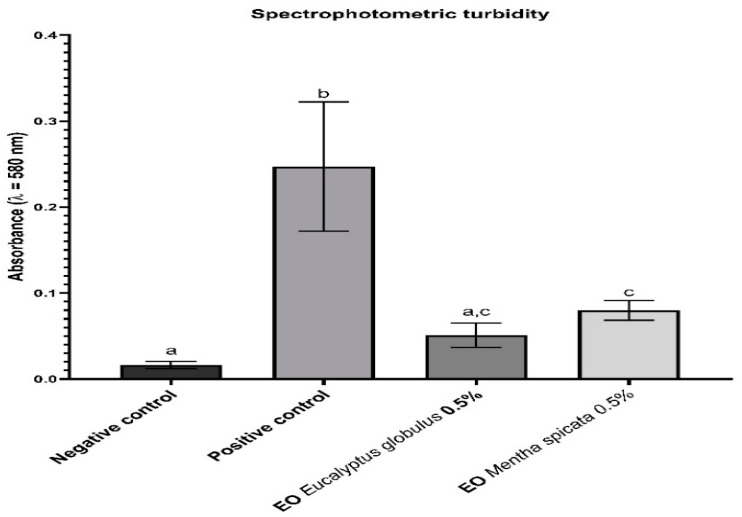
Spectrophotometric turbidity.

**Figure 4 antibiotics-12-00369-f004:**
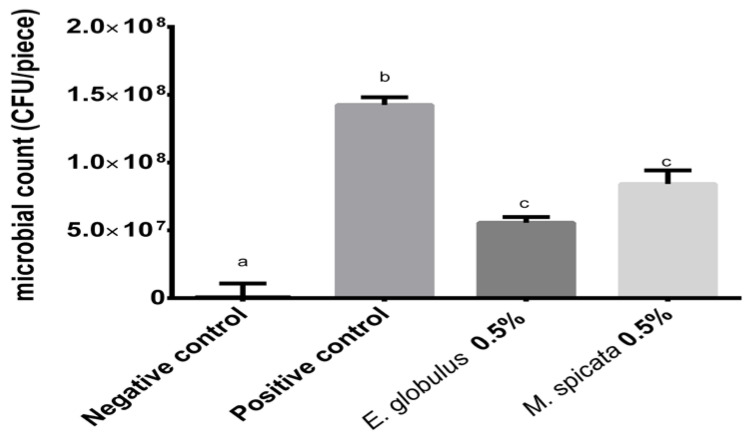
Recoverable microorganism count. The letters indicate which groups had a statistically significant difference with a confidence interval above 95%.

**Table 1 antibiotics-12-00369-t001:** Components in the essential oil of *M. spicata*.

N	Compound	Retention Index	% of Relative Area
1	α-Pinene	936	3.77
2	Sabinene	956	1.16
3	α-Myrcene	985	2.00
5	3-Carene	1010	3.70
6	Limonene	1020	12.91
7	1,8-Cineol (eucalyptol)	1033	1.20
8	γ-Terpinene	1062	0.33
9	cis-α-Terpineol	1078	0.20
10	Terpinolene	1086	0.41
11	*p*-Menthan-3-one	1092	0.70
12	Menthone	1125	0.37
13	1-menthol	1136	1.88
14	Carvone	1142	57.93
15	2-isopropyl-5-methyl-3-cyclohexen-1-one	1156	0.20
16	*p*-Menthane	1263	5.88
17	Copaene	1376	1.44
18	α-Bourbonene	1385	0.61
19	Caryophyllene	1420	1.03

**Table 2 antibiotics-12-00369-t002:** Components in the essential oil of *E. globulus*.

N	Compound.	Retention Index	% of Relative Areas
1	α-Pinene	936	18.15
2	β-Pinene	983	1.78
3	α–Myrcene	985	1.56
4	1,8-Cineol (eucalyptol)	1033	65.83
5	4-Terpineol	1182	0.77
6	α-Terpinol	1242	2.02
7	α-Terpineol acetate	1330	5.14
8	(+)-Aromadendrene	1440	2.84
9	Globulol	1590	1.92

**Table 3 antibiotics-12-00369-t003:** Inhibitory activity of the essential oils of *M. spicata* and *E. globulus*.

Essential Oil	Concentration	Diameter ± SD (mm)
*Mentha spicata*	100%	18.30 ± 0.47
	50%	15.75 ± 0.41
	10%	13.66 ± 0.47
*Eucalyptus globulus*	100%	27.00 ± 0.82
	50%	18.00 ± 0.82
	10%	12.00 ± 0.82
Solvent (DMSO)	0.1%	0

## Data Availability

The datasets generated during and/or analyzed during the current study are available from the corresponding author on reasonable request.

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
