# Peer review of "Inhibitory Activity of Essential Oils of Mentha spicata and Eucalyptus globulus on Biofilms of Streptococcus mutans in an In Vitro Model"

_antibiotics, 2023, doi:10.3390/antibiotics12020369_

Round 1

Reviewer 1 Report

Abstract

Line 27-29: You indicated that the EOs used were commercially available… My question is as follows: is information on their composition not available? does the supplier not provide this information? If he provided, why redo these analyses?

Lines 31-33 : what is the meaning of “y”  between 1.8484 mg/mL and 1.9168 mg/mL?

Introduction

Lines 49-52: kindly support your Sentences with at least one reference.  The same for lines 88-89

Results

I assume that you did not have a standard for each compound presented in tables 1 and 2, and that you used available data to identify these compounds. It is mandatory to add reference for each compound.

Lines 116-117: remove the capital letters.

Line 198: replace the figure 4 with another one with better resolution.

Materials and methods

The section 4.2 should be better described and the way you identified each compound should be mentioned.

Section 4.3.1: why you did not supplement this part with the crystal violet attachment assay? 

Lines 439-440: where are the Supplementary Materials?

Author Response

REVIEWER 1

Line 27-29: You indicated that the EOs used were commercially available… My question is as follows: is information on their composition not available? does the supplier not provide this information? If he provided,

 why redo these analyses?

R1: Thank you for your observations, its composition is not available by the Peruvian company in spite of being one the main manufacturer of essential oil in Peru. For this reason, we re-analyzed by GC-masas and confirm its fingerprint of each EO and according to our results, we confirmed the identity of the samples.

Lines 31-33: what is the meaning of “y” between 1.8484 mg/mL and 1.9168 mg/mL?

R2: I am sorry, it was deleted and replaced by “and”.

Introduction

Lines 49-52: kindly support your Sentences with at least one reference.  The same for lines 88-89

R3: Thank you for this detail, we included new references.

Results

I assume that you did not have a standard for each compound presented in tables 1 and 2, and that you used available data to identify these compounds. It is mandatory to add reference for each compound.

R4: Thank you for this detail, we re-write the methodology and added “ Each retention index (RI) was calculated compared with a homologous series of n-alkanes C9–C25 (C9, BHD purity 99% and C10–C25, Fluka purity 99%). The relative amount (expressed as a percentage) of each compound identified in the EO was calculated by comparing the area of the corresponding peak in the chromatogram with the total area of identified peaks. No correction factor was applied.”

Lines 116-117: remove the capital letters.

R5: Thank you for this detail, we removed the capital letters.

Line 198: replace the figure 4 with another one with better resolution.

R6: Thank you for this detail, we included a new figure for 4.

Materials and methods

The section 4.2 should be better described and the way you identified each compound should be mentioned.

R7: Thank you for your observation, we re-write the methodology.

Section 4.3.1: why you did not supplement this part with the crystal violet attachment assay?

R8: Than you for you observation, in effect we could have worked with crystal violet but our method used Congo red according to previous studies and biofilm formation by bacteria also is studied in vitro by testing either in microtiter plates or on Congo red agar (CRA).

  1. Seo M, Oh T, Bae S. Antibiofilm activity of silver nanoparticles against biofilm forming Staphylococcus pseudintermedius isolated from dogs with otitis externa. Vet Med Sci. 2021 Sep;7(5):1551-1557. doi: 10.1002/vms3.554. Epub 2021 Jun 22. PMID: 34156766; PMCID: PMC8464246.
  2. Zaidi S, Singh SL, Khan AU. Exploring antibiofilm potential of bacitracin against streptococcus mutans. Microb Pathog. 2020 Dec;149:104279. doi: 10.1016/j.micpath.2020.104279. Epub 2020 Jun 5. PMID: 32512154.

Lines 439-440: where are the Supplementary Materials?

R9: Thank you for your observation, we included a supplementary material (Figure S1).

Reviewer 2 Report

Page 2 : lines 64-75. Why do authors include this information in the introduction. Authors use a single species batch model. There is no reason to explore other models unless authors make a statement on the basis of which arguments they have chosen their current model.

Page 3, Table 1 and Table 2: Does the accuracy of the GC-MS analysis allow for three digits to be mentioned. When presenting 3 digits authors claim that the GC-MS analysis has an accuracy of 0.001% relative area, is that correct?

Page 4 Table 1 and Page 5, Figure 1 : It looks as if the Solvent (G in Figure 1) also shows a halo but in Table 3 the diameter is stated as 0 mm. Can authors comment on this?

Page 4 / Page 5 : treatment of biofilms with EO’s.

In the materials & methods section it is stated that at t=72h the medium was retired by absorption for the last time. But the data on pH level at t=72h is lacking from figure 2. Why did authors not analyse the spend medium at t=72h for its pH?

In the M&M section(page 12 first paragraph) the procedure for the treatment of the biofilms is described. The treatment solutions also contained polysorbate, a surfactant. Authors tested the surfactant for its antimicrobial effect in a separate not reported test. But in the biofilm model the surfactant might also just remove the biofilm without killing the bacteria. Did the authors test the potential removal of the biofilm by the surfactant? The effectivity of the EO’s is determined by analyzing the amount of biofilm that is left behind or killed.

When the biofilm is treated with 0.5% of the EO’s the test solution is removed after 1 minute and then the slabs are rinsed with 0.9% sterile saline. Is that a single rinse with 0.9% saline? Or is it a repeated rinse with sterile saline? When a single rinse is applied some of the EO will be left behind (as it is difficult to completely remove the liquid from the well) and then the experiment will be run at a sub-MIC dose with the respective EO’s. Can authors comment on this?

Authors perform plate counting on the remaining biofilms in the treatment experiment. Plate counting is a more accurate procedure than measuring turbidity. Why do authors still report turbidity as well?

The data derived from  M&M section 4.3.1. seems to be lacking from the results section. Or did I not understand the procedure by which the inhibition halos were obtained?

Discussion.

On page 8 lines 259-267 the data on the MIC values is discussed. Authors state that the data is according to Rasooli et al who reported a MIC value of 2x 10-3 mg/mL. But that is factor 1000 below the values reported in this study (1.84 and 1.92 mg/mL respectively). Is there a typo in any of the reported values?

Do authors consider a reduction of bacterial counts of less than 1 log a biological significant effect? I do agree that the results are statistically significant.

Authors suggest that the biofilm might become resistant to the EO’s (Page 9, line 290-291). Is there any proof (in literature) that bacteria actually become resistant against these EO’s?

Author Response

REVIEWER 2

Page 2 : lines 64-75. Why do authors include this information in the introduction. Authors use a single species batch model. There is no reason to explore other models unless authors make a statement on the basis of which arguments, they have chosen their current model.

R1: Thank you for your observations, we modified this paragraph according to your suggestions.

Page 3, Table 1 and Table 2: Does the accuracy of the GC-MS analysis allow for three digits to be mentioned. When presenting 3 digits authors claim that the GC-MS analysis has an accuracy of 0.001% relative area, is that correct?,

R2: thank you for your observations, in effect according to your suggestion, we only included two decimals in our table as well as the retention index of each compound.

Page 4 Table 1 and Page 5, Figure 1 : It looks as if the Solvent (G in Figure 1) also shows a halo but in Table 3 the diameter is stated as 0 mm. Can authors comment on this?

R3: There is no halo, but there is a coloration by diffusion of the solvent used as a control.

Page 4 / Page 5 : treatment of biofilms with EO’s.

In the materials & methods section it is stated that at t=72h the medium was retired by absorption for the last time. But the data on pH level at t=72h is lacking from figure 2. Why did authors not analyse the spend medium at t=72h for its pH?

R4: Than you for your observation, actually each read is given at 72h of growth, but the pH measurements are given when the medium is renewed, which are given on days 2 and 3 (that is why there are 6 readings). it should be noted that the influence of the treatment can be observed from the second pH measurement, since the first dose of treatment was carried out after the first 24 hours of culture. Likewise, it is visible that the impact of the treatment on the decrease in pH only has a relevant influence in the first two measurements (22.5 h and 26.5 h), after 46 hours of culture the difference between the groups is no longer so marked.

In the M&M section (page 12 first paragraph) the procedure for the treatment of the biofilms is described. The treatment solutions also contained polysorbate, a surfactant. Authors tested the surfactant for its antimicrobial effect in a separate not reported test. But in the biofilm model the surfactant might also just remove the biofilm without killing the bacteria. Did the authors test the potential removal of the biofilm by the surfactant? The effectivity of the EO’s is determined by analyzing the amount of biofilm that is left behind or killed.

R5: It is determined by the amount of biofilm that remains adhered and the count is made, therefore it is viable.

When the biofilm is treated with 0.5% of the EO’s the test solution is removed after 1 minute and then the slabs are rinsed with 0.9% sterile saline. Is that a single rinse with 0.9% saline? Or is it a repeated rinse with sterile saline? When a single rinse is applied some of the EO will be left behind (as it is difficult to completely remove the liquid from the well) and then the experiment will be run at a sub-MIC dose with the respective EO’s. Can authors comment on this?

R6: Thank you for your detail, it was not a single rinse, it is a triple rinse.

Authors perform plate counting on the remaining biofilms in the treatment experiment. Plate counting is a more accurate procedure than measuring turbidity. Why do authors still report turbidity as well?

R7: Actually it is to reinforce our results, the count is always probable to variations due to pipetting, dilutions, etc... and we consider it important to reinforce with the reading by equipment.

The data derived from M&M section 4.3.1. seems to be lacking from the results section. Or did I not understand the procedure by which the inhibition halos were obtained?

R8: Thank you for your observation, we included this paragraph: According to our results based on the evaluation of the ability to form biofilms by the Congo Red method were observed punctate, dry and black colonies. Clearly showing that the black pigmentation fades through the agar medium (Figure S1, Supplementary material). This supports the ability of S. mutans ATCC 25175 to form biofilm in tooth. This shows that S mutans is a biofilm former, somewhat tacit because dental plaque is the best-known biofilm model.

Discussion.

On page 8 lines 259-267 the data on the MIC values is discussed. Authors state that the data is according to Rasooli et al who reported a MIC value of 2x 10-3 mg/mL. But that is factor 1000 below the values reported in this study (1.84 and 1.92 mg/mL respectively). Is there a typo in any of the reported values?

R9: Thank you for this observation, we removed 10-3 because we wrote incorrect this value and was modified in text according to the article used as reference.

Do authors consider a reduction of bacterial counts of less than 1 log a biological significant effect? I do agree that the results are statistically significant.

R10: In effect, we reported as statistically significant to the EOs compared to positive control but comparing both EOs, they have similar results and there is no statistical significant.

Authors suggest that the biofilm might become resistant to the EO’s (Page 9, line 290-291). Is there any proof (in literature) that bacteria actually become resistant against these EO’s?

R11: Thank you for your observation, in effect we did not have any reference to confirm the resistant of essential oil against S. mutants.

Round 2

Reviewer 1 Report

Thank you for your corrections and answers to comments. All the issue raised have been addressed. 

Good luck and best regards.

Author Response

Dear reviewer, thank you for your comments.

Reviewer 2 Report

Authors have adressed the issues raised by me in the first review. 

There is 1 issue that authors adress insufficiently:

In the M&M section (page 12 first paragraph) the procedure for the treatment of the biofilms is described. The treatment solutions also contained polysorbate, a surfactant. Authors tested the surfactant for its antimicrobial effect in a separate not reported test. But in the biofilm model the surfactant might also just remove the biofilm without killing the bacteria. Did the authors test the potential removal of the biofilm by the surfactant? The effectivity of the EO’s is determined by analyzing the amount of biofilm that is left behind or killed.

R5: It is determined by the amount of biofilm that remains adhered and the count is made, therefore it is viable.

Authors lack to understand that a treatment with polysorbate (the solvent for the EO's) would be an essential control in this study. Authors cannot distinguish the effect of the solvent and the EO's when the solvent is not tested. The killing efficacy of polysorbate in planktonic cultures  was tested. But not the effect of polysorbate on attached biofilms. If Polysorbate was able to remove some of the biofilm the effect of the EO's would be overestimated.

Author Response

In the M&M section (page 12 first paragraph) the procedure for the treatment of the biofilms is described. The treatment solutions also contained polysorbate, a surfactant. Authors tested the surfactant for its antimicrobial effect in a separate not reported test. But in the biofilm model the surfactant might also just remove the biofilm without killing the bacteria. Did the authors test the potential removal of the biofilm by the surfactant? The effectivity of the EO’s is determined by analyzing the amount of biofilm that is left behind or killed.

Authors lack to understand that a treatment with polysorbate (the solvent for the EO's) would be an essential control in this study. Authors cannot distinguish the effect of the solvent and the EO's when the solvent is not tested. The killing efficacy of polysorbate in planktonic cultures was tested. But not the effect of polysorbate on attached biofilms. If Polysorbate was able to remove some of the biofilm the effect of the EO's would be overestimated.

R1: Dear reviewer, thank you very much for your observation. We are sorry that our answer was not adequate to clarify this question. However, we do carry out the test on the activity of 1% polysorbate 20 and 0.01% NaCl on biofilm, and for this reason we have clarified it in page 12, lines (418-420) and it says: a positive control (inoculated and under treatment only with control vehicle), a negative control (Not inoculated and under treatment only with control vehicle), as presented in Figures 3 and 4, those figures show values of both controls.

Round 3

Reviewer 2 Report

Thanks for clarifying my comment. Now I understand your data better.